# RETRACTED: *In Vitro* Induction of Apoptosis in Isolated Acute Myeloid Leukemia Cells: The Role of *Anastatica hierochuntica* Methanolic Extract

**DOI:** 10.3390/metabo12090878

**Published:** 2022-09-17

**Authors:** Islam M. El-Garawani, Amira S. Abd El-Gaber, Noura A. Algamdi, Aamer Saeed, Chao Zhao, Omar M. Khattab, Mohamed F. AlAjmi, Zhiming Guo, Shaden A. M. Khalifa, Hesham R. El-Seedi

**Affiliations:** 1Department of Zoology, Faculty of Science, Menoufia University, Shebin El-Kom 32512, Egypt; 2Department of Chemistry, Faculty of Science, Menoufia University, Shebin El-Kom 32512, Egypt; mero_as92@yahoo.com (A.S.A.E.-G.); omarkhattab500@gmail.com (O.M.K.); 3Botany Department, Faculty of Science, Taibah University, Al Madinah 42353, Saudi Arabia; nagamdi@taibahu.edu.sa; 4Department of Chemistry, Quaid-i-Azam University, Islamabad 45320, Pakistan; asaeed@qau.edu.pk; 5College of Food Science, Fujian Agriculture and Forestry University, Fuzhou 350002, China; zhchao@live.cn; 6Pharmacognosy Group, Department of Pharmaceutical Biosciences, Uppsala University, Biomedical Centre, SE 751 24 Uppsala, Sweden; 7Department of Pharmacognosy, College of Pharmacy, King Saud University, Riyadh 11451, Saudi Arabia; malajmii@ksu.edu.sa; 8School of Food and Biological Engineering, Jiangsu University, Zhenjiang 212013, China; guozhiming@ujs.edu.cn; 9Department of Molecular Biosciences, The Wenner-Gren Institute, Stockholm University, SE 106 91 Stockholm, Sweden; shaden.khalifa@regionstockholm.se; 10International Joint Research Laboratory of Intelligent Agriculture and Agri-Products Processing, Jiangsu Education Department, Jiangsu University, Nanjing 210024, China; 11International Research Center for Food Nutrition and Safety, Jiangsu University, Zhenjiang 212013, China

**Keywords:** acute myeloid leukemia blasts, anticancer, ^1^H-NMR, LC-ESI-MS, Brassicaceae, molecular networking, metabolites

## Abstract

*Anastatica hierochuntica* L. (Cruciferae) has been known in Egyptian folk medicine as a remedy for gastrointestinal disorders, diabetes and heart diseases. Despite the wide usage, *A. hierochuntica* research provides insufficient data to support its traditional practice. The cytotoxicity of *A. hierochuntica* methanolic extract was investigated on acute myeloid leukemia blasts (AML) and normal human peripheral leucocytes (NHPL). The phytochemical identification of bioactive compounds using ^1^H-NMR and LC-ESI-MS was also performed. *A. hierochuntica* extract caused non-significant cytotoxicity on NHPL, while the cytotoxicity on AML was significant (IC_50_: 0.38 ± 0.02 μg/mL). The negative expression of p53, upregulation of Caspase-3 and increase in the BAX/BCL-2 ratio were reported at the protein and mRNA levels. The results suggest that *A. hierochuntica* extract induced AML cell death via the p53-independent mitochondrial intrinsic pathway and further attention should be paid to this plant as a promising natural anticancer agent.

## 1. Introduction

*Anastatica hierochuntica* (L.), also known as Kaff Maryam, is an aromatic plant in the family Brassicaceae (Cruciferae). It is a well-known medicinal plant, grown in the deserts and used in folk medicine for the treatment of various disorders, including spasms, uterine hemorrhage and fatigue [1,2]. It is also used for the treatment of gastrointestinal disorders, high blood pressure, fever, malaria, epilepsy, fatigue, diabetes, heart diseases and infertility [2,3]. Two novel skeletal flavanones, anastatins A and B, were isolated from *A. hierochuntica*, in addition to seven known flavonoids, eleven aromatic compounds, three phenyl propanoids, twelve lignanes and four flavono-lignans [4,5]. Quercetin, isovitexin and 3-rutinoside of kaempferol and quercetin were also identified from *A. hierochuntica* [6,7]. Twenty compounds, including a series of flavone *C*- and *O*- glycosides, flavonols, phenolic acids and chlorogenic acids, were reported in herbal tea made from an aqueous extract of *A. hierochuntica* seeds [8].

Recent studies on *A. hierochuntica* extracts showed their potent bioactivity, such as a potent therapeutic effect against drug-induced nephrotoxicity, especially in the aqueous extract, which has the potential capability to restore oxidative stability and improve kidney function after CCl_4_ acute kidney injury [9]. Moreover, *A. hierochuntica* aqueous extract administration revealed a significant reduction in the corrected maternal weight gain of dams and the body weight of fetuses compared to controls [10]. The antiproliferative effect through the induction of apoptosis in MCF-7 cells was proven following incubation with *A. hierochuntica* extracts [11].

Leukemia is a devastating human cancer that usually results in abnormally high numbers of white blood cells, regardless of its type and classification [12]. Global epidemiological studies have demonstrated that the incidence and mortality still rank highly in regard to the worldwide population. South Korea has the highest incidence-to-mortality ratio in both sexes, followed by Cyprus and Canada [13]. The current treatments for leukemia depend on chemotherapy and immunotherapy. Such treatments have considerable adverse effects. Hence, the need for a better cure for leukemia has been highlighted recently and natural products represent an enriched resource of novel bioactive components. Some natural extracts have antioxidant properties [14,15] that lead to the termination of the chemically induced genotoxicity and cytotoxicity *in vivo* [16], and *in vitro* [17]. Others have anticancer potential *in vivo* and *in vitro* [18,19,20]. In continuation of our project to identify new, pharmacologically active sources from Egyptian medicinal plants [21,22,23,24], we investigated, for the first time, the flavonoid-rich methanolic extract of *A. hierochuntica*. Additionally, we performed detailed molecular networking through the LC-ESI-MS platform and the presence of chemical classes was confirmed by ^1^H-NMR. The *in vitro* anticancer potential of *A. hierochuntica* extract on isolated acute myeloid leukemia (AML) blasts in comparison to its effect on isolated normal human peripheral leucocytes was assessed for the first time on the extract compared to other new and relevant studies [11].

## 2. Materials and Methods

### 2.1. Materials

#### 2.1.1. Plant Material and Extraction

Dried aerial parts of *A. hierochuntica* L. were purchased from Nobi El-Attar, a commercial supplier in Menoufia, Egypt. They were identified by a taxonomist, Prof. Dr. Zaki Turki (Department of Botany, Faculty of Science, Menoufia University, Egypt). A voucher specimen (No. 1000) was deposited in the herbarium of the same department.

Two kilograms of *A. hierochuntica* dried aerial parts were ground and defatted one time with hexane (60–80 °C) for one week at room temperature and extracted two times under reflux with 70% methanol/water (6 L each for 3 days). The methanolic extract was evaporated under reduced pressure using a rotary evaporator (BUCHI, Switzerland) in order to produce 50 g residue. The extract was subjected to ^1^H-NMR and LC-ESI-MS to explore its constituents and screen the chemical structure. For biological screening, the extract was dissolved in DMSO with a final concentration of no more than 0.1% (*v/v*) in culture media.

#### 2.1.2. Cisplatin

Cisplatin [Pt(NH_3)2_Cl_2_], UNISTIN^®^, EIMC United Pharmaceuticals, Egypt, was utilized in this study as a positive control, with a final concentration of 3.0 µg/mL in culture media.

### 2.2. Methods

#### 2.2.1. Phytochemical Analysis

##### ^1^H-NMR Analysis

^1^H-NMR spectra were recorded on a Jeol EX-400 spectrometer: 400 MHz. HSQC spectra were recorded at 298 K on a Bruker 600 MHz (TCI CRPHe TR-1H and 19F/13C/15N 5 mm-EZ CryoProbe) spectrometer. Chemical shifts were referenced to the solvent peak for (CD_3_)_2_SO at δH 2.50 ppm and δ^13^C 39.52 ppm.

##### LC-MS-MS Analysis

The methanolic extract was dissolved in 10% acetonitrile (AcN) to a final concentration of 5 mg/mL and filtered through a 0.45 mm membrane filter (Corning^®^, Wiesbaden, Germany). LC-MS and LC-MS/MS analyses were carried out on a Waters CapLC coupled to a Waters Q-Tof Micro, equipped with a LockSpray interface. Leucine enkephalin (*m*/*z* 556.2771) was used as lockmass for the LC-MS analysis. The mobile phase consisted of solvent A (95% water, 5% AcN, 0.1% formic acid) and solvent B (5% water, 95% AcN, 0.1% formic acid). Separations were performed using a Phenomenex Jupiter C18 column (150 × 21.2 mm) and the following gradient: 0–3 min isocratic at 95:5 (A:B), 3–47 min linear gradient from 95:5 (A:B) to 5:95 (A:B) and 47–48 min linear gradient from 5:95 (A:B) to 95:5 (A:B) and finally equilibrated with 95:5 (A:B) in 48–60 min. The flow rate was 40 mL/min. A sample injection volume of 10 mL was employed. Data were processed using MassLynx version 4.1. S [25,26,27].

Using the Global Natural Products (GNPS) Social Molecular Networking dataset, the raw MS file was analyzed. By analyzing the similarity between the fragmentation pattern from the raw mass spectrum and the GNPS library, GNPS assists in the identification and discovery of metabolites. Other installed programs, including MSConvert, File Zilla and Cytoscape version 3.5.1, were used to operate with GNPS [28,29]. A fixed collision energy of 1.4 kV was used to acquire the QTOF-(+)MS/MS data, which were subsequently sent to the GNPS server after being converted from RAW to mzXML file format using MSConvert [30,31].

#### 2.2.2. *In Vitro* Anticancer Investigations

##### Peripheral Blood Leucocyte Isolation and Incubation

The extract’s toxicity versus tolerance assessment was performed on normal human peripheral leucocytes (NHPL) isolated from seven healthy non-smoker volunteers. Moreover, the anticancer properties were evaluated on peripheral venous blood samples of seven acute myeloid leukemic (AML) male patients who were recently diagnosed and had not yet received chemotherapeutic treatment (before samples were taken). Around two milliliters of peripheral venous blood was collected using sterile syringes and then transferred to sterile K_2_-EDTA tubes (KEMICO vacutainer, Cairo, Egypt). All samples were transported to the laboratory within two to three hours. The study plan was approved by the Menoufia University ethical committee (MUFS-F-GE-5-20) and followed the Institutional Ethical Committee guidelines at the Faculty of Science, Menoufia University, Egypt.

Both NHPL and AML cells were mixed with five folds of erythrocyte lysing buffer (0.015 M NH_4_C1, 1 mM NaHCO_3_, 0.l mM EDTA). Then, they were centrifuged for 5 min at 1000 rpm using a cooling centrifuge (Sigma 3K 30, Osterode, Germany). These steps were repeated until a white pellet appeared [32]. The extract was added to the isolated NHPL and AML cells, which were cultured in serum-free RPMI-1640 medium supplemented with 1% (100 U/mL penicillin and 100 μg/mL streptomycin) for 3 h and incubated at 37 °C in a humid 5% CO_2_ atmosphere [33].

##### Trypan Blue Assay

The *in vitro* cytotoxicity assay was carried out using charged cationic trypan blue dye (0.4%, Lonza, Switzerland). The minimal inhibitory concentration (IC_50_) of *A. hierochuntica* extract on AML cells was obtained from cytotoxicity percentages against the serial concentration curve (0, 0.5, 1.0, 3.0, 6.0 and 10 μg/mL). To evaluate the mechanism of action, the investigated NHPL and AML cells were incubated with 0, 0.5 and 3.0 μg/mL of *A. hierochuntica* (70% MeOH) extract for three hours at 37 °C in a humidified 5% CO_2_ atmosphere. Cisplatin (3.0 μg/mL) was used as a positive control.

##### Acridine Orange/Ethidium Bromide (AO/EB) Dual-Fluorescence Staining

Cell viability was confirmed in NHPL and AML by one-minute incubation of the cell suspension with 10% of (1:1) AO/EB (10 µg/mL) dye mixture on a clean glass slide. Then, cells were immediately examined by a fluorescent microscope (Olympus BX 41, Tokyo, Japan) at 400× magnification. Two types of cells were identified, based on the fluorescence emission: viable bright green-colored cells with intact structure and late with intact structure and late apoptotic or dead cells with orange to red color. Approximately 1000 cells per slide were evaluated [34].

##### Annexin-V/Propidium Iodide (PI) Dual-Fluorescence Staining of AML

In order to qualitatively assess cell death stages and mechanisms, one microliter of Annexin-V/PI (100 µg/mL) mixture was added to nine microliters of treated AML cell suspension on a clean glass slide, and then cells were immediately examined by a fluorescent microscope (Olympus BX 41, Tokyo, Japan) at 400× magnification to distinguish the stages of cell death. Early apoptosis (stained positive for Annexin-V and negative for PI), late apoptosis or cell death (stained positive for both Annexin-V and PI) and necrosis (stained positive for PI) could be observed; viable cells were not visible (stained negative for both Annexin-V and PI). Approximately 1000 cells were evaluated per slide.

##### Immunocytochemical Study of AML Cells

After incubation, AML cells were smeared on positively charged slides and then processed for p53, Caspase-3, BAX and BCL-2 immunocytochemical protein levels using an avidin–biotin complex immunoperoxidase technique [35]. All proteins were detected using specific anti-human monoclonal antibodies (Dako, Glostrup, Denmark). Fields were chosen randomly and cells were imaged. At least five fields per stained slide were scored. The mean percentage of positive cells in all groups was used as an immunocytochemical scoring system (positive cells/total number of counted cells) × 100. Cells were examined (400×) using a light microscope (Olympus BX 41, Tokyo, Japan).

##### Quantitative Real-Time Polymerase Chain Reaction

In order to assess the expression of apoptosis-related genes, the *p53*, *CASP-3*, *BCL-2* and *BAX* mRNA was determined. Total RNA was isolated from AML cells using an RNeasy Plus Minikit (Thermo Fisher Scientific, Austin, TX, USA). Then, the synthesis of complementary DNA (cDNA) was carried out using the Revert Aid™ H Minus Reverse Transcriptase (Fermentas, Thermo Fisher Scientific Inc., Sunnyvale, CA, USA). Real-time PCR reactions were performed using Power SYBR^®^ Green (Thermo Fisher Scientific, Austin, TX, USA) on an Applied Biosystems 7500 system. The values of gene expression were normalized to Glyceraldehyde-3-phosphate dehydrogenase (*GAPDH*) as a housekeeping gene. Data on primers used in this study are provided in Table 1**.**

#### 2.2.3. Statistical Analysis

All experiments were performed three times. Data are presented as (mean±SD) and were analyzed by either one-way ANOVA or Student’s *t*-test. Figures were drawn using Microsoft Excel 2019. The probability level *p* < 0.05 was considered statistically significant. The statistical analyses were conducted using SPSS (version 25, IBM, New York, NY, USA).

## 3. Results

### 3.1. UPLC-MS-QTOF and Molecular Networking Analysis

Several studies have found that natural flavonoids exert growth-inhibitory effects on various types of cancer cells. Flavonoids have the ability to easily bind to cell membranes, penetrate *in vitro* cultured cells and modulate cellular metabolic activities. Flavonoids’ anticarcinogenic activities include oxidative damage reduction, carcinogen inactivation, inhibition of proliferation, promotion of differentiation, induction of cell cycle arrest and apoptosis, impairment of tumor angiogenesis and suppression of metastasis [36,37].

In the present study, LC-ESI-MS analysis was performed for the methanolic extract of *A. hierochuntica* using the positive ion mode technique. Four compounds were identified as flavones and another one as a flavonol, which were previously reported [38]. Molecular networking was applied in this study for further peak selection and identification. All the nodes are reflections of metabolites with unique peaks in the raw mass spectra, and the tentative identification of the compounds is shown in Table 2. Compound **L1** is a flavonoid conjugated with glucose and rhamnose units. It showed a molecular ion peak at *m*/*z* 611 [M+H]^+^, which gave a fragment at *m*/*z* 449 due to the loss of glucose unit [M-162+H]^+^, identified tentatively as luteolin-*O*-glucoside-*O*-rhamnoside. **L2** was tentatively identified as diosmetin-*C*-glucoside-*O*-pentoside, with a molecular ion peak at *m*/*z* 597 and fragment ion at *m*/*z* 463 [M-(pentose)+H] ^+^. **L3** was tentatively identified as apigenin-*C*-glucoside-*O*-glucoside at *m*/*z* 594, which showed a molecular ion peak at *m*/*z* 433 due to the loss of glucose [M-(glucose)+H]^+^. **L4** showed a molecular ion peak at *m*/*z* 597 and two fragments at *m*/*z*: 449 [M-(rhamnose)+H] ^+^, 287 [M-rhamnose-glucose+H] ^+^, suggesting that **L4** is kaempferol-*O*-glucoside-*O*-rhamnoside. Finally, **L5** showed a molecular ion peak at *m*/*z* 609 and fragment ion at *m*/*z* 433 [M-(glucuronic acid)+H] ^+^ and was tentatively identified as apigenin-*O*-glucoside-*O*-glucuronide. Compounds **L5**–**L14** are flavonoids that were annotated using the GNPS library (Figure 1 and Figure 2).

As described in Table 2, quercetin (**L8**, **L10**, **L13**) appears to be a molecule with multiple properties, all of which are aimed at reducing cell growth in cancer cells. In conclusion, quercetin sensitizes synergistically induced cell death in human malignant cell lines to CD95/TRAIL and increases apoptosis in B cells [50]. The apoptotic effects of luteolin (**L1**, **L6**) on HL-60 cells were found to be dose-dependent. Luteolin’s apoptotic effects were also time-dependent. Flow cytometry was also utilized to investigate the impact of 100 μM luteolin on the apoptotic ratio of HL-60 cells over time. The apoptotic effects of luteolin were also time-dependent, with the percentage of apoptotic HL-60 cells increasing dramatically from 3 h (3.2%) to 6 h (33.6%) with 60 μM luteolin treatment [51]. Kaempferol (**L4**, **L9**) also exhibited antiproliferative and proapoptotic effects on human leukemia cells by inhibiting Akt-mediated pro-survival cascades, increasing the intracellular BAX/BCL2 ratio and decreasing the expression of MDR-associated genes [52]. Apigenin (**L3**, **L5**, **L14**), on the other hand, was found to cause a dose-dependent reduction in viable leukemia cells. Apigenin’s IC_50_ on HL60 was 30 μM, with complete viability loss at 100 μM. In contrast, erythroleukemic cells were more resistant, with both K562 and TF1 cell lines showing a 66% reduction in viability at 100 μM [53].

### 3.2. Assessment of Methanolic Extract’s ^1^H-NMR Spectra

The ^1^H-NMR spectra obtained for the *A. hierochuntica* methanolic extract showed a dominance of signals in the aliphatic region (e.g., methyl proton peak at *δ* 0.9, methylene proton (CH_2_)_n_ peaks in the region at *δ* 1–3.5 ppm). There are particular carbohydrates and glycosides (e.g., anomeric hydrogen at *δ* 5.2 ppm), in addition to some aromatic moieties at region *δ* (6.2–7.8 ppm); see Figure 3. The heteronuclear single quantum coherence spectroscopy (HSQC) decoupled sensitive spectrum showed the presence of a large amount of glycosides as follows: CH groups as a red spot at a range of *δ* 60–80 ppm and SP2 methylene groups with blue-colored spots at a range of *δ* 54–66 ppm. Additionally, anomeric protons’ CH groups were red spots around a chemical shift at *δ* 100 ppm. Apigenin and luteolin are flavanones characterized by the presence of a singlet proton at H-3 at *δ* 6.9 ppm, with ^13^C at 107 ppm. Apigenin, naringenin and kaempferol are characterized by a 1,4 di-substitution benzene ring H-2′-3′ at *δ* 7.48 and 6.94 and ^13^C *δ* 129 and 116 ppm, respectively (Figure 4) [54].

### 3.3. Assessment of A. hierochuntica’s Effect on NHPL and AML Viability

The investigated normal and AML leucocytes were treated with the crude extract from the aerial parts of the *A. hierochuntica* plant and then processed with trypan blue; see Figure 5. The extract induced dose-dependent cytotoxicity (IC_50_: 0.38 ± 0.02 μg/mL) in AML cells against serial concentrations. The extract induced an increase in the cytotoxicity by approximately 9.5 and 10.4-folds for A1 and A2, respectively, when compared with untreated cells. However, acridine orange/ethidium bromide dual-fluorescence staining revealed the same pattern of cytotoxicity, with approximately 5.8 and 9.1-folds for A1 and A2, respectively, when compared with untreated cells; see Figure 6. Otherwise, no significant toxicity was observed with NHPL, except for the cisplatin-treated cells, which showed significant (*p* ˂ 0.05) toxicity when using both staining methods.

### 3.4. Qualitative Determination of Cell Death Stages by Annexin-V/PI Dual-Fluorescent Staining in AML Cells

In order to assess the mode of cell death, AML cells were stained, after 3 h of incubation with various treatments, with Annexin-V/PI, and the results revealed that dead cells were stained with an orange to red color, providing a qualitative assessment of necrotic or late apoptotic stages of cell death; see Figure 6.

### 3.5. Immunocytochemical Reactivity of Apoptosis-Related Proteins in AML Cells

The protein levels of p53, Caspase-3 and the BAX/BCL-2 ratio were detected by immunocytochemistry after 3 h of various treatments; see Figure 7. With respect to untreated cells, the percentage of positively stained AML cells revealed a statistically significant (*p* ˂ 0.05) increase in the levels of Caspase-3 protein of approximately 2.5 and 3.2-folds for A1 and A2, respectively. The elevated values of the BAX/BCl-2 ratio were also significant, with approximately 3.2- and 3.7-fold increases for A1 and A2, respectively. Otherwise, the tumor suppressor protein, p53, showed non-significant values (*p* ˂ 0.05) in both control and treated AML cells, except for the cisplatin-treated groups, which showed statistically significant (*p* ˂ 0.05) positive values (68.3 ± 2.89) compared to controls (2.0 ± 1.00); see Figure 7 and Figure 8.

### 3.6. The mRNA Expression of Apoptosis-Related Genes in AML Cells

The effect of *A. hierochuntica* treatment on p53, CASP-3 and BAX/BCL-2 mRNA expression levels using qRT-PCR was investigated. *A. hierochuntica* altered the CASP-3 gene expression and proportionally altered the BAX/BCL-2 ratio in AML cells after 3 h of treatment; see Figure 9. In comparison to control cells, CASP-3 was significantly upregulated in the *A. hierochuntica* (0.5 µg/mL) and cisplatin groups. A higher concentration of the extract (3.0 µg/mL) showed a lower value of upregulation. Moreover, the BAX/BCL-2 ratio had a significant elevation with *A. hierochuntica* treatment when compared to the untreated controls, with approximately 44 and 69% for A1 and A2, respectively. However, the p53 expression was unchanged when compared to the solvent (DMSO) and positive control (cisplatin) groups. These results suggested that *A. hierochuntica* treatment altered the expression of apoptosis-related genes, inducing mitochondrial intrinsic apoptosis in AML cells independently of p53 expression.

## 4. Discussion

In the treatment of cancers such as AML, resistance to cytotoxic chemotherapy becomes a major obstacle [55]. The natural extracts of *A. hierochuntica* have great antioxidant and antitumor effects [56]. Our results revealed that the crude extract was rich in glucosidic flavonoids (luteolin and apigenin), possessing structures that exhibit C2=C3 double bonds—see Figure 10—which were previously reported for their anticancer potential [40] and are in agreement with earlier information regarding the impact of flavonoids and other antioxidants [39]. Further, Table 2 illustrates several identified compounds with reported potency as anti-inflammatory and antioxidant [39], cytotoxic [40], antimicrobial [42] and anti-helminthic [43] agents.

Taking into account the great value of dietary flavonoids of the *A. hierochuntica* plant, it is appealing to consider their complementary role as antioxidants and cancer-preventive agents [47]. Similarly, the anticancer impact of the majority of natural derivatives was proven to exhibit selective cytotoxicity towards cancerous cells, with non-significant toxicity towards normal cells [21,23]. Consistent with Al-Eisawi et al. [57], the 70% MeOH extract treatments revealed a non-significant cytotoxic effect on isolated NHPL, whereas it had a 73.0 ± 4.12% cytotoxic effect on AML cells.

The two primary modes of cell death are apoptosis and necrosis. Annexin-V/PI staining was performed to investigate the type of cell death that was induced, and the late apoptotic effect of the extract was displayed [30]. The AO/EB fluorescent staining also quantitatively confirmed the late apoptotic and direct necrotic cell death, as indicated by the presence of a structurally normal orange nucleus [14,31]. The AO stain can cross the plasma membrane of viable and early apoptotic cells, while EB is only taken up by cells with weakened cytoplasmic membrane integrity [35]. On the other hand, trypan blue stains all stages of apoptosis and necrosis. Variable AML cytotoxicity results between trypan blue and AO/EB fluorescent staining were obtained. The higher dose of the extract induced higher cytotoxicity, and this may explain the lower values of Caspase-3 expression with a higher dose of the treatment due to the increased internal cellular dysfunction and, eventually, the elevated cell death rate. In addition, the higher concentration of the extract caused increased toxicity and stress, leading to the induction of apoptosis via the Caspase-independent pathway [57,58,59]. This may be caused by several processes, such as autophagy, mitotic catastrophe or endoplasmic reticulum stress [60].

The elevated BAX/BCL-2 ratio, as well as the lack of p53 protein expression, in this study suggested a possible cell death cascade via a mitochondrial, p53-independent pathway and cytochrome c release [61]. The presence of a C2=C3 double bond in luteolin and apigenin was correlated in earlier studies with mitochondrial damage and cell death pathways [62], in addition to the evidential toxicity of the leukemia cell lines [40]. The *A. hierochuntica* plant is rich in flavonoids and flavonoid glycosides, as confirmed by the phytochemical screening in this study, which is consistent with its anticancer potential [40].

In apoptosis, BAX proteins activate the cascade of mitochondrial reactions via the release of cytochrome c, and the successive activation of Caspase-3 ultimately leads to apoptosis [61]. The mitochondrial pathway is crucially important for apoptotic induction. Mitochondrial dysfunction is attributed to an increase in the BAX/BCL-2 ratio and Caspase-3 activation as the two major apoptotic key players [63,64,65,66]. In AML cells, mitochondria exhibit high respiratory activity while having a lower coupling efficiency, which is associated with elevated proton leakage and lower spare reserve capacity. These are likely to explain the increased responsiveness of AML cells towards mitochondria-targeted drugs. This indicates that mitochondria can be an efficient and selective therapeutic target [32,37,38,39]; however, the intrinsic apoptotic pathway can also be induced through the negative regulation of the anti-apoptotic protein expression of BCL-2 [11]. In the current study, *A. hierochuntica* elevated Caspase-3 expression and the ratio of BAX/BCL-2. Otherwise, p53 showed no significant expression, suggesting that the extract induced p53-independent mitochondrial intrinsic apoptosis in AML cells.

## 5. Conclusions

The present work reports, for the first time, to the best of our knowledge, the *in vitro* anticancer activity of *A. hierochuntica* methanolic extract against acute myeloid leukemia blasts. The extract exhibited a selective cytotoxic impact on AML, through the mitochondrial intrinsic apoptotic pathway, while it did not exert any cytotoxicity towards the normal human peripheral leucocytes *in vitro*. Further studies are warranted to screen the gene expression regulation in blood and other cancer cell types using *A. hierochuntica* extracts/compounds.

## Figures and Tables

**Figure 1 metabolites-12-00878-f001:** *A. hierochuntica* metabolite parent mass molecular network. The blue circular nodes refer to all metabolite parent masses revealed from raw mass spectra. The yellow ones are for the metabolites that were identified by the GNPS library. Seven of the flavonoids were verified by literature comparison, and the remaining ones were manually identified (**L1**–**L14**).

**Figure 2 metabolites-12-00878-f002:** Mirror image of identified compounds from methanolic extract of *A. hierochuntica* compared to spectra of GNPS library.

**Figure 3 metabolites-12-00878-f003:** ^1^H-NMR spectrum of *A. hierochuntica* methanolic extract.

**Figure 4 metabolites-12-00878-f004:** HSQC spectrum of *A. hierochuntica* methanolic extract.

**Figure 5 metabolites-12-00878-f005:** *A. hierochuntica* cytotoxicity on NHPL and AML cells after 3 h of incubation with different doses, as detected by trypan blue (TB) exclusion method. Modes and stages of cell death were detected by acridine orange/ethidium bromide dual-fluorescent staining (AO/EB). Data represent means ± SD of three independent experiments (n = 3). NHPL: normal human peripheral leucocytes; AML: acute myeloid leukemia; TB: trypan blue. *: significant with respect to control (*p* ˂ 0.05). A1: *A. hierochuntica* methanolic extract (0.5 µg/mL); A2: (3.0 µg/mL); cisplatin: (3.0 µg/mL).

**Figure 6 metabolites-12-00878-f006:** Representative photomicrographs of control and treated AML cells after 3 h of incubation, as observed by an inverted microscope and detected by acridine orange/ethidium bromide (AO/EB) dual-fluorescent staining. Assessment of necrotic or late apoptotic cell death as detected by Annexin-V/Propidium.

**Figure 7 metabolites-12-00878-f007:** Representative photomicrographs of control and treated AML cells after 3 h of incubation. Protein levels as evaluated by immunocytochemical reactivity (brown staining); negative p53 (blue staining) was observed in control and treated groups, and positive reactivity with brown staining development in the cisplatin group. Decreased BCL-2 protein immunoreactivity was observed, whereas elevated levels of BAX and Caspase-3 (CASP-3) protein expression were noticed with different treatments. A1: *A. hierochuntica* methanolic extract (0.5 µg/mL); A2: (3.0 µg/mL); Cis: cisplatin (3.0 µg/mL).

**Figure 8 metabolites-12-00878-f008:** The percentage of immunocytochemical positive AML cells in control and treated groups after 3 h of incubation. Data represent the average of three independent experiments for the protein expression of p53, CASP-3 and BAX/BCL-2 ratio (n = 3). Bars: standard deviation; *: significant with respect to the control (*p* ˂ 0.05). CASP-3: Caspase-3; A1: *A. hierochuntica* methanolic extract (0.5 µg/mL); A2: (3.0 µg/mL); cisplatin: (3.0 µg/mL).

**Figure 9 metabolites-12-00878-f009:** The fold change of *p53*, *CASP-3* and *BAX/BCL-2* ratio of control and treated AML cells’ mRNA expression after 3 h of incubation. Data represent the average of three independent experiments (n = 3). Bars: standard deviation. Columns with different letters are statistically significant (*p* ˂ 0.05). *CASP-3*: Caspase-3; A1: *A. hierochuntica* methanolic extract (0.5 µg/mL); A2: (3.0 µg/mL); cisplatin: (3.0 µg/mL).

**Figure 10 metabolites-12-00878-f010:** Chemical structure of glucosidic flavonoids (luteolin and apigenin).

**Table 1 metabolites-12-00878-t001:** Primer sequences of genes analyzed by real-time PCR.

Name	Accession Number	Sense (5′–3′)	Antisense (5′–3′)
*GAPDH*	NM_001289745.2	GGATTTGGTCGTATTGGG	GGAAGATGGTGATGGGATT
*P53*	NM 001289746.1	GGATTTGGTCGTATTGGG	GGAAGATGGTGATGGGATT
*BCL-2*	NM_001114735.1	TACAGGCTGGCTCAGGACTAT	CGCAACATTTTGTAGCACTCTG
*BAX*	NM_001291431.1	CCCGAGAGGTCTTTTTCCGAG	CCAGCCCATGATGGTTCTGAT
*Caspase-3*	NM_001354777.1	GGCGCTCTGGTTTTCGTTAAT	CAGTTCTGTACCACGGCAGG

Abbreviations: *GAPDH*: glyceraldehyde-3-phosphate dehydrogenase; *BCL-2*: B-cell lymphoma 2; *BAX*: *BCL-2*-associated X protein.

**Table 2 metabolites-12-00878-t002:** Compounds (**L1**–**L14**) identified from *A. hierochuntica*.

Peak No.	RT (min.)	[M+H]^+^	*m*/*z* Fragment	Tentative Identification	GNPS Link	Literature Review of the Biological Activities of the Compounds
**L1**	20.92	611.24	270.97, 286.95, 449.05	Luteolin-*O*-glucoside-*O*-rhamoside	Literature	Anti-inflammatory and antioxidant [39]; cytotoxic [40]
**L2**	21.95	597.48	301.18, 463.53	Diosmetin-*C*-glucoside-*O*-pentoside	Literature	DPPH activity [41]
**L3**	22.93	594.54	433.024, 271.13	Apigenin-*C*-glucoside-*O*-glucoside	Literature	Antimicrobial activity [42]; antioxidant [39]; cytotoxic [40]
**L4**	24.591	597.39	449.31, 287.14	Kaempferol-*O*-glucoside-*O*-rhamnoside	Literature	Anti-leishmania [43]
**L5**	27.617	609.24	433.026, 270.97	Apigenin-*O*-glucoside-*O*-glucuronide	Literature	Anti-neurodegenerative and antioxidant [44]; cytotoxic [40]
**L6**	16.8	449.31	153.05, 287.009	Luteolin-*O*-glucoside	http://gnps.ucsd.edu/ProteoSAFe/gnpslibraryspectrum.jsp?SpectrumID=CCMSLIB00003136499 (accessed on 25 July 2022)	Anticancer potency (inhibition of melanogenesis and proliferation in B16 melanoma 4A5 cells) [4,44,45,46,47,48]
**L7**	21.4	273.25	152.95, 146.96	Naringenin	http://gnps.ucsd.edu/ProteoSAFe/gnpslibraryspectrum.jsp?SpectrumID=CCMSLIB00003136767 (accessed on 25 July 2022)
**L8**	19.6	303.19	132.1, 153.001, 229.02, 257.02, 275.09, 285.03	Quercetin	http://gnps.ucsd.edu/ProteoSAFe/gnpslibraryspectrum.jsp?SpectrumID=CCMSLIB00000205749 (accessed on 25 July 2022)
**L9**	14.2	287.27	135, 52.9, 164.9, 177, 213, 227.1, 241, 259.1	Kaempferol	http://gnps.ucsd.edu/ProteoSAFe/gnpslibraryspectrum.jsp?SpectrumID=CCMSLIB00005720284 (accessed on 25 July 2022)
**L10**	12.8	627.22	287.97, 302.98, 465.01	Quercetin-*O*-diglucoside	Literature
**L11**	21.65	303.18	286.01, 272.99, 144.99, 152.99, 177	Diosmetin	Literature	DPPH activity [41]
**L12**	17.85	465.37	165.10, 229.12, 257.08, 303.03	Hyperoside	http://gnps.ucsd.edu/ProteoSAFe/gnpslibraryspectrum.jsp?SpectrumID=CCMSLIB00000213809 (accessed on 25 July 2022)	Inhibition of melanogenesis and proliferation in B16 melanoma 4A5 cells [47]
**L13**	15.8	463.36	300.97	Isoquercitin	http://gnps.ucsd.edu/ProteoSAFe/gnpslibraryspectrum.jsp?SpectrumID=CCMSLIB00003136664 (accessed on 25 July 2022)
**L14**	21.7	433.476	119.10, 153.06, 271.13	Apigenin-*O*-glucoside	http://gnps.ucsd.edu/ProteoSAFe/gnpslibraryspectrum.jsp?SpectrumID=CCMSLIB00000222717 (accessed on 25 July 2022)	[38,46,48,49]

## Data Availability

The data presented in this study is available in article.

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
