# Peer review of "RETRACTED: In Vitro Induction of Apoptosis in Isolated Acute Myeloid Leukemia Cells: The Role of Anastatica hierochuntica Methanolic Extract"

_metabolites, 2022, doi:10.3390/metabo12090878_

Round 1

Reviewer 1 Report

·       The authors neglected to write the Molecular Networking procedure, either in the experimental or discussion sections. Also, they focused to annotate the flavonoid constituent only. What about the other characterized structures such as coumarins and glucosinolates which are specific to the investigated species?

·       Please add the GNPS links.

·       No need to replicate the keywords included in the title, Brassicaceae and Molecular networking could be used.

·       In Table 1, Please union all fragments with or without decimal.

·       The figure’s quality needs to be improved, for example, the lettering is too small in Fig. 1, 2.  They should be clearly readable and of high quality.

·       Finally, please cite specific relevant references which help the author in the identification of metabolites.

Author Response

Response letter enclosed

Reviewer 2 Report

Although A. hierochuntica is a plant widely used since ancient times, studies of this plant are still scarce and its biological activities and reputed medical benefits require further investigation. Previous studies have shown that the extracts of this plant have pharmacologically important anticancer properties, such as for the treatment of breast cancer, but there are still no studies on acute myeloid leukemia. Therefore, this study adds important information about the potential of this plant. The topic falls in the scope of this journal. I have read the manuscript with interest and I found this manuscript very complete, detailed and well written. The design is reasonable and the authors have given good number of citations about the subject. Therefore, I consider the manuscript suitable for publication.

There are only small changes to be made: 

Lines 84-85 – Add how much methanol/water (70%) was used for each extraction.

Line 110 - Please check and correct the column data Phenomenex Jupiter C18 column (150 1 mm i.d., 5 mm).

Line 221 - Add a space in “at δ 1”.

Line 277 - Add a space before the Figure 8.

Lines 284-294 - Write not in italics.

Lines 284-285-287-290-293-353 – Write “A. hierochuntica” in italics.

Line 394 - Correct the wordhirerochuntica”.

Line 363 - Add end point.

Author Response

Response letter enclosed

Reviewer 3 Report

The paper evaluated the effect of methanol extract of A. hierochuntica over acute myeloid leukemia (AML) blasts.  Some chemical constituents, especially flavonoids, were annotated from GNPS platform through LC-ESI-MS data and the presence of chemical classes was confirmed by 1H NMR data. The crude extract showed cytotoxicity on AML, inducing cell death via the p53-independent mitochondrial intrinsic pathway.  Two minor corrections should be incorporated into the text:

1. Line 75: biological studies instead biological students

2. Figure 2: from my point of view this figure could be transferred to supplementary material.

Author Response

Response letter enclosed

Reviewer 4 Report

The manuscript describes and discusses logically designed experiments and presents results that are expected to be of large interest for the scientific community. It is an interesting study with an interesting approach. The paper in the whole is well designed and results sound. Nevertheless, the manuscript needs a minor revision:

Point 1: In the introduction part should be more highlighted the main aim of the paper, and additionally, what is the novelty of carried research work.

Point 2: How do the Authors select the analytes? The rational of the choice of the selected biologically active compounds studied is missing and should be clearly discussed.

Point 3: Quality of the figures must be improved.

Author Response

Response letter enclosed.
